# A Multi-Strategy Improved Sparrow Search Algorithm for Coverage Optimization in a WSN

**DOI:** 10.3390/s23084124

**Published:** 2023-04-20

**Authors:** Hui Chen, Xu Wang, Bin Ge, Tian Zhang, Zihang Zhu

**Affiliations:** School of Computer Science and Engineering, Anhui University of Science and Technology, Huainan 232001, China; wangxu100110@163.com (X.W.);

**Keywords:** wireless sensor network, coverage optimization, sparrow search algorithm, non-dominated sorting, two-sample learning strategy

## Abstract

To address the problems of low monitoring area coverage rate and the long moving distance of nodes in the process of coverage optimization in wireless sensor networks (WSNs), a multi-strategy improved sparrow search algorithm for coverage optimization in a WSN (IM-DTSSA) is proposed. Firstly, Delaunay triangulation is used to locate the uncovered areas in the network and optimize the initial population of the IM-DTSSA algorithm, which can improve the convergence speed and search accuracy of the algorithm. Secondly, the quality and quantity of the explorer population in the sparrow search algorithm are optimized by the non-dominated sorting algorithm, which can improve the global search capability of the algorithm. Finally, a two-sample learning strategy is used to improve the follower position update formula and to improve the ability of the algorithm to jump out of the local optimum. Simulation results show that the coverage rate of the IM-DTSSA algorithm is increased by 6.74%, 5.04% and 3.42% compared to the three other algorithms. The average moving distance of nodes is reduced by 7.93 m, 3.97 m, and 3.09 m, respectively. The results mean that the IM-DTSSA algorithm can effectively balance the coverage rate of the target area and the moving distance of nodes.

## 1. Introduction

### 1.1. Background of Problem

Wireless sensor networks (WSNs) are now widely used in many fields such as environment monitoring, industrial control, military safety, and health care, and are playing a crucial role in the development and application of IoT technology. However, in some special environments such as forests, deserts, and oceans, the sensor nodes are randomly deployed in the target monitoring area, which leads to coverage vulnerabilities and coverage redundancy in the networks, resulting in poor network performance and wasted resources. The coverage problem is one of the most fundamental issues in wireless sensor networks, which determines the quality of service of wireless sensor networks. A reasonable and effective node deployment strategy can not only reduce the network cost but can also improve the network efficiency to a large extent. Improving the coverage rate of networks by deploying mobile nodes has become a mainstream coverage optimization method, but the method greatly increases the network deployment cost [1,2,3]. Therefore, how to control the moving distance of mobile nodes while improving the coverage rate of networks is of great significance for the future development of the entire wireless sensor network application.

### 1.2. Contributions

The main contributions of this paper are as follows:A multi-strategy improved sparrow search algorithm for WSN coverage is proposed.In order to optimize the initial population, this paper introduces the Delaunay triangulation to search for the uncovered areas of the networks.The global search capability of the algorithm is improved by introducing the idea of non-dominated sorting to optimize the population of the explorer.To prevent the algorithm from easily falling into the local optimum in the later stages, a two-sample learning strategy is used to improve the follower position update method.The algorithm is compared with three other WSN optimization algorithms. It is proved to be effective not only in improving the network coverage, but also in reducing the moving distance of sensor nodes.

### 1.3. Organization

The rest of this paper is organized as follows. Section 2 describes the node coverage model of WSNs. Section 3 describes the original sparrow search algorithm. Section 4 describes in detail the proposed multi-strategy improved sparrow search algorithm for WSN coverage optimization. Section 5 details the experimental scheme and the comparison of experimental results for coverage optimization. Section 6 concludes the work of this paper.

## 2. Related Works

At present, scholars at home and abroad have conducted a lot of research on the coverage optimization problem of wireless sensor networks in planar areas, and they have made some progress. The main research methods are divided into two types: one is coverage optimization based on geometric methods [4,5,6] and the other is coverage optimization based on intelligent algorithms. Mahboubi et al. [7] proposed a node deployment algorithm based on Voronoi diagrams, where the virtual forces exerted by the vertices and boundaries of polygons are used to find the new locations of nodes, and the simulation results prove that the algorithm can improve the coverage rate. Liu et al. [8] optimized node deployment by an improved virtual force algorithm, which considered the interaction forces between nodes based on the traditional virtual force algorithm, and the interaction forces between nodes could drive the sensor nodes to cover the whole monitoring area, but it was still difficult to achieve a high coverage rate in the end. Wang et al. [9] combined the improved virtual force algorithm with the improved gray wolf algorithm, which can effectively make the nodes more evenly distributed and reduce the average travel distance of the nodes, but the complexity of this algorithm is high and requires a long running time.

In recent years, with the development and wide application of intelligent algorithms, researchers have found that the problems of WSN coverage optimization can be effectively solved by intelligent algorithms, such as particle swarm algorithm, whale algorithm, firefly algorithm, and gray wolf algorithm [10,11]. Intelligent algorithms have the advantages of simple computation and strong search capability. Mahnaz et al. [12] proposed an improved whale optimization algorithm for WSN coverage optimization to solve the complex coverage problems by developing the approaches of exploration, spiral attack, and bubble net attack. Experimental results show that the algorithm can extend the life cycle of the network, but it is easy for the algorithm to fall into local optimum. Kavita et al. [13] proposed a node deployment method based on the gray wolf algorithm, which maximizes coverage while ensuring network connectivity by improving the fitness function and the population location update method, but the algorithm does not take into account the cost caused by node movement. Chao et al. [14] proposed a node deployment method based on the improved artificial swarm algorithm and the teaching strategy. The simulation results show that the algorithm achieves a better balance between global search and local search, but the complexity of the algorithm is high. Aparajita et al. [15] combined the firefly algorithm with the Voronoi diagram and K-means algorithm and calculated the optimal sensing radius by Voronoi cell structure, and then found the optimal deployment location of nodes by the firefly algorithm. The simulation results showed that the algorithm could improve the coverage and lifetime of wireless sensor networks. Wang et al. [16] proposed an adaptive, discrete-space-oriented wolfpack optimization algorithm for movable wireless sensor networks. First, an adaptive expansion strategy based on a minimum overlapping full-coverage model was designed to achieve the minimum overlap and gap-free coverage of the monitoring area. Secondly, the target-node probability matrix and adaptive step size are improved. The simulation results show that the algorithm has better convergence speed and global optimization capability. Zhu et al. [17] proposed a coverage optimization method based on an improved weed algorithm. The different stages of the weed algorithm are improved to enhance the search capability of the algorithm as well as the convergence speed of the algorithm. Teng et al. [18] proposed a novel hybrid firework-virtual force algorithm based on the µ-Lawwhich, which improves the search ability and accuracy of the algorithm through the idea of μ-law in non-uniform quantization and introduces the virtual force algorithm to accelerate the evolution of the fireworks population and improve the efficiency of the algorithm, but the algorithm does not optimize the moving distance of nodes in the process of network coverage optimization.

Yin et al. [19] proposed a WSN coverage optimization method based on the Yin-Yang pigeon-inspired optimization algorithm, which improved coverage ratio and convergence. However, the algorithm is not efficient and cannot be applied in complex environments. Wu et al. [20] proposed a multi-objective optimization algorithm for WSNs based on improved particle swarm optimization, which aims at solving the problems of coverage hole and redundancy. Simulation results indicate that this algorithm can improve coverage ratio and reduce the moving distance of nodes. However, the effect of the algorithm is unstable. He et al. [21] proposed a WSN coverage optimization model based on an improved marine predator algorithm (IMPA). The simulation results demonstrate that the IMPA has a better coverage rate than other metaheuristic algorithms. However, the algorithm does not pay attention to the energy consumption of nodes. Huang et al. [22] proposed a node coverage optimization strategy with an improved COOT bird algorithm (COOTCLCO). Simulation results show that COOTCLCO has a faster convergence speed and better search accuracy, but the algorithm only considers the coverage rate of the networks.

Wang et al. [23] used an improved sparrow search algorithm to improve the node coverage of wireless sensor networks, and the improved algorithm had some improvement in search accuracy and convergence speed, but it ignored the impact of the location distribution of the sparrow population in the initial stage on the performance of the algorithm. Duan et al. [24] proposed an improved sparrow search algorithm to optimize the coverage of wireless sensor networks, introduced the initialized population based on the good point set in response to the uncertainty generated by the standard sparrow search algorithm using random initialized population, and also proposed evaluation metrics for coverage issues. The results show that combining the good point set to initialize the initial position of the population can effectively improve the convergence speed and accuracy of the algorithm, but the study does not fully consider the impact of the moving distance of mobile nodes on the cost of network coverage optimization in the optimization process. Wu et al. [25] proposed a multi-objective coverage optimization method based on the improved sparrow search algorithm, which not only improves the network coverage by improving the sparrow search algorithm but also prevents some nodes from moving for too long, and the results show that the algorithm can balance the network coverage and node movement cost to a certain extent, but the method of controlling the moving distance of sensor nodes by this algorithm is still inadequate and fails to reduce the movement distance of all nodes as a whole.

Recent research on the coverage optimization of wireless sensor networks has yielded some results, but there are still some problems. Firstly, most studies only consider how to improve the coverage of the network, ignoring the fact that there is a large amount of energy consumption during the movement of the nodes. Secondly, in recent years, only a small number of researchers have conducted coverage optimization studies for networks where hybrid nodes are deployed. Static nodes cannot move to other places after being deployed, but they are less expensive than the mobile nodes. Although the sparrow search algorithm has successfully solved some problems in the field of WSNs, it is still difficult to effectively solve the multi-objective optimization problem. In order to effectively balance the coverage rate of networks and the moving distance of sensor nodes in the coverage optimization, a multi-strategy improved sparrow search algorithm for WSN coverage optimization is proposed.

## 3. WSN Node Coverage Model

In this paper, we set up a WSN in which all sensor nodes have the same structure and properties, and each sensor node has a defined sensing radius R and communication radius $R_{c}$. To ensure the connectivity of the network, the communication radius of a node is usually set to be two times the node’s sensing radius. Assume that the set of wireless sensor nodes is $S=\left\{s_{1},s_{2},s_{3},\ldots ,s_{n}\right\}$, and the coordinates of any sensor node $s_{i}$ in the set of nodes can be expressed as $\left(x_{i},y_{i}\right)$. The set of monitoring points is $M=\left\{m_{1},m_{2},m_{3},\ldots ,m_{n}\right\}$, the coordinates of any monitoring point $m_{j}$ in the monitoring area can be expressed as $\left(x_{j},y_{j}\right)$. The Euclidean distance between a sensor node and a monitoring node in the monitoring area is:(1)$$d\left(s_{i},m_{j}\right)=\sqrt{{\left(x_{i}-x_{j}\right)}^{2}+{\left(y_{i}-y_{j}\right)}^{2}}$$

The node-sensing model in this paper is a Boolean sensing model [26], then the probability that the target point $m_{j}$ to be monitored by the sensor node $s_{i}$ is set as follows:(2)$$p_{cov}\left(s_{i},m_{j}\right)=\{\begin{array}{c} 1,\;\;\;if\;d\left(s_{i},m_{j}\right)\leq R\;\\ 0,\;\;\;\;\;\;\;\;\;otherwise \end{array}$$

The joint sensing probability of all sensor nodes to the monitoring points $m_{j}$ in the monitoring area is:(3)$$C_{p}\left(s_{all},m_{j}\right)=1-\overset{n}{\underset{i=1}{\prod }}\left[1-p_{cov}\left(s_{i},m_{j}\right)\right]$$
where $s_{all}$ is all sensor nodes in the monitoring area. Assuming that the monitoring area is a rectangle and A×B denotes the size of the monitoring area. The rectangular monitoring area is divided into A×B identical grids of equal size, and the monitoring point m is located at the center of the grid. According to the above equation, the joint sensing probability of monitoring points can be calculated, and the coverage area is obtained after accumulation. The calculation formula of coverage is as follows:(4)$$C_{r}=\sum_{x=1}^{A}\sum_{y=1}^{B}\frac{C_{p}\left(s_{all},m_{\left(x-1\right)B+y}\right)}{AB}$$

Suppose the initial position of node *i* in the region is $\left(x_{i}^{ini},y_{i}^{ini}\right)$,the target location is $\left(x_{i}^{fin},y_{i}^{fin}\right)$, then the sum of the moving distances of all nodes in *S* is as follows:(5)$$d\left(s\right)=\overset{n}{\underset{i=1}{\sum }}\sqrt{{\left(x_{i}^{fin}-x_{i}^{ini}\right)}^{2}+{\left(y_{i}^{fin}-y_{i}^{ini}\right)}^{2}}$$

According to the above analysis, the objective function is g(x). The calculation formula of g(x) can be expressed as follows:(6)$$g\left(x\right)=\varphi C_{r}+\omega \frac{1}{d\left(s\right)}$$
where φ and ω are the weighting factors, and φ+ω=1.

## 4. Standard Sparrow Search Algorithm

In the sparrow search algorithm (SSA) [27,28], the population is divided into two main parts: the explorer and the follower. Among them, the explorer is responsible for finding food and guiding the whole population to the foraging area, while the follower acquires food according to the position of the explorer. The explorer’s position is updated in the following way:(7)$$X_{i,j}^{t+1}=\{\begin{array}{c} X_{i,j}^{t}\cdot exp\left(\frac{-i}{\alpha \cdot Iter_{max}}\right),R_{2}<ST\\ X_{i,j}^{t}+Q\cdot L,\;\;\;\;\;\;\;\;\;\;R_{2}\geq ST \end{array}$$
where *t* indicates current iteration, $Iter_{max}$ is a constant with the largest number of iterations. α∈(0,1] is a random number, Q is a random number which obeys normal distribution. L represents a matrix of 1×*n* for which each element is 1. $R_{2}\in \left[0,1\right]$ and ST∈[0.5,1] represent the alarm value and safety threshold, respectively. When $R_{2}<ST$, which means that no predator is found around the current foraging environment, the explorer can enter the wide search mode. If $R_{2}\geq ST$, it means that some sparrows have discovered the predator and issued an alarm—all the sparrow needs to quickly move to the safety position.

The updated equation for the followers is as follows:(8)$$X_{i,j}^{t+1}=\{\begin{array}{c} Q\cdot exp\left(\frac{X_{worst}-X_{i,j}^{t}}{i^{2}}\right),i>\frac{N}{2}\\ X_{p}^{t+1}+\left|X_{i,j}^{t}-X_{p}^{t+1}\right|\cdot A^{+}\cdot L,otherwise \end{array}$$
where $X_{p}$ and $X_{worst}$ are the current global best and worst positions. *A* represents a matrix of 1×*n* for which each element is randomly assigned 1 or −1, and $A^{+}=A^{T}{\left(AA^{T}\right)}^{-1}$. When i>N/2, it suggests that the *i*th sparrow with worse fitness values is most likely to be starving.

During the foraging process of the sparrow population, 10% to 20% of the sparrows will act as defenders, and when danger approaches, all sparrows will abandon their current food and move to a new location, and the defenders will be updated as follows:(9)$$X_{i,j}^{t+1}=\{\begin{array}{c} X_{best}^{t}+\beta \left|X_{i,j}^{t}-X_{best}^{t}\right|,f_{i}>f_{best}\\ X_{i,j}^{t}+K\left(\frac{\left|X_{i,j}^{t}-X_{worst}^{t}\right|}{\left(f_{i}-f_{worst}\right)+\varepsilon }\right),f_{i}=f_{best} \end{array}$$
where $X_{best}$ is the current global optimal location. β is a random number that obeys the standard normal distribution. K∈[−1,1] is a random number. $f_{i}$ is the fitness value of the present sparrow, $\;f_{best}$, $f_{worst}$ represent the current best and worst fitness values, ε is the smallest constant to avoid the denominator being 0.

## 5. Multi-Strategy Improved Sparrow Search Algorithm

The initial sparrow search algorithm solves problems by searching and updating explorers, followers, and defenders. It has high search accuracy, fast convergence, and good stability compared with intelligent algorithms such as particle swarm algorithm and ant colony algorithm. However, its application in multi-objective optimization problems is not yet mature. In the later stage of the algorithm, there is the problem of the gradual reduction of diversity. Therefore, this paper proposes a multi-strategy improved sparrow search algorithm for WSN coverage optimization. Firstly, the uncovered areas are searched by the Delaunay triangulation strategy and screened as the initial population of the sparrow search algorithm; secondly, the idea of non-dominated sorting is introduced to improve the quality of the explorer population, which helps to alleviate the problem that it is difficult to balance multiple objectives with the algorithm; and finally, a two-sample learning strategy is used to optimize the updated formula for the followers, which avoids the problem that it is easy to fall into local optimum in the later stage of the algorithm.

### 5.1. Initial Population Optimization

In bionic intelligence algorithms such as SSA, particle swarm optimization (PSO), and ant colony optimization (ACO), in order to search for the solutions to the problem in the early stages of the algorithm, they usually use random initialization to determine the initial population. Currently, researchers mainly determine the initial population by using chaotic mappings and other random methods, which shows that the initial population is important to the overall performance of the algorithm. It can affect the global search effect as well as the convergence speed of the algorithm. In this paper, we introduce the Delaunay triangulation strategy [11] to locate the approximate locations of coverage vulnerabilities. These locations are then filtered and used as the initial population for the coverage optimization algorithm in this paper.

After dividing the static nodes in the monitoring area by Delaunay triangulation strategy, the triangle network is shown in Figure 1a. The new triangular network is shown in Figure 1b after adding the boundary points of the monitoring area to the triangular network. In the network constituted by all static nodes, the uncovered area is mainly located inside the triangle, and the relationship between the Delaunay triangle and uncovered area is divided into three kinds. When max(d(A,B),d(A,C),d(B,C))<2R, there are no uncovered areas inside the triangle. When max(d(A,B),d(A,C),d(B,C))=2R, there exists an uncovered region inside the triangle, located entirely inside the triangle, only when the triangle is an acute triangle. When max(d(A,B),d(A,C),d(B,C))>2R, if this triangle is an acute or right triangle, there must be an uncovered area inside the triangle; if this triangle is an obtuse triangle and the vertical distance from point *C* to side *AB* is greater than the sensing radius of the sensor node, then there is also an uncovered area inside the triangle [11]. Where the *d(A,B)* represents the distance between point *A* and point *B*, and the R represents the sensing radius of a sensor node. 

The approximate location of coverage vulnerabilities in the network can be quickly located based on the relationship between Delaunay triangles and uncovered areas. Considering the significant difference in the size of coverage vulnerabilities existing in the network, the location of the initial population needs to be further filtered. The sensing radius of sensor nodes is fixed and the same, and the size of the coverage hole inside the triangle is related to the position of the three vertices of the triangle. The relationship is analyzed as follows:

The area of the coverage vulnerability is related to the area of the triangle, the area of the sector and the overlapping areas between the sectors, respectively. The schematic is shown in Figure 2. Since the sum of the interior angles of a triangle is 180 degrees, the sum of the areas of the sectors inside any triangle is the same. The area of the triangle can be calculated from the coordinate positions of the three nodes. The area of the overlapping areas in the sector is related to the distance between the nodes and the sensing radius of the nodes. Define the variable *Cl* to measure the size of the coverage hole inside each triangle with the following formula:(10)$$Cl=S_{\Delta }-S_{st}+S_{ol}$$
where $S_{\Delta }$ is the area of the triangle, $S_{st}$ is the area of the sector in the triangle, $S_{ol}$ is the area of the overlapping area in the triangle. $S_{ol}$ is calculated as follows:(11)$$S_{ol}=\overset{3}{\underset{i=1}{\sum }}\frac{\pi R^{2}}{180}\cdot arcos\frac{d_{i}}{2R}-\frac{d_{i}}{2}\sqrt{R^{2}-\frac{d_{i}{}^{2}}{4}}$$
where $d_{i}$ represents the distance between two sensor nodes, *R* is the sensing radius of the nodes.

The triangles can be sorted according to the value of *Cl*, and the top *N* positions of the centroids of triangles are selected as the initial population. *N* is the population size of the IM-DTSSA algorithm.

### 5.2. Explorer Population Optimization

In the standard SSA algorithm, explorers usually consist of higher energy population individuals, which are responsible for searching food-rich areas in the whole population, providing search guidance for followers, and playing a crucial role in the search effectiveness of the whole population. Therefore, both the quality and the number of explorer populations have a significant impact on the overall performance of the algorithm. The better the quality and larger the number of explorer populations, the better the global search ability of the algorithm, but it will slow down the convergence speed, and only a good balance between the quality and number of explorers in the SSA algorithm can make the performance of the algorithm better [29].

The energy level of an individual depends on the fitness value of the individual, and how to evaluate the fitness value in the process of multi-objective optimization is an urgent problem to be solved. In this paper, we introduce the idea of non-dominated sorting [30] to sort the sparrow populations, and secondly further sort them by calculating the entropy of crowding distance of individuals in the same non-dominated rank. In NSGA, a solution x is said to dominate y when its value is better than another solution y for all fitness functions. All sparrows are divided into different non-dominated fronts using the non-dominated rank, and the schematic diagram is shown in Figure 3. It can be easily seen that the number of solutions that dominate *A* or *B* is 0, so solution *A* and solution *B* are divided into the first non-dominated front. It is obvious that both solution *A* and solution *B* are able to dominate solution *C*, so solution *C* is divided into the second non-dominated front.

Define $f_{1}\left(X\right)=\frac{1}{cr\left(X\right)}$, cr(X) is the coverage rate of the network when the nodes are deployed according to the scheme of solution *X*. Define $f_{2}\left(X\right)=amd\left(X\right)$, amd(X) is the average moving distance of the mobile nodes when the nodes are deployed according to the scheme of solution *X*. 

Set the population as P and calculate the two parameters n(p) and S(p) for each individual p. Where n(p) is the number of individuals in the population that dominate p and S(p) is the set of individuals in the population that are dominated by p. The fast non-dominated sorting process is as follows:(1)Find all individuals of the population with n(p) = 0 and deposit them in the set F(1);(2)For the current individual *i* in F(1), the set S(*i*) consists of the individuals dominated by *i*. Iterate through all individuals k in the set S(*i*), performing n(k) = n(k) − 1. If n(k) = 0, deposit individual k in the set F(2).(3)All individuals in F(1) form the first layer of non-dominated surfaces. With F(2) as the current set, the above operation is repeated until all the individuals are deposited in the corresponding set.

The calculation of the crowding distance entropy CD(*i*) of each solution is used in the NSGA algorithm to measure the equilibrium and diffusion of solutions. It can help analyze the denseness of solutions on the same non-dominated fronts. However, this paper also hopes that the solution that can balance the coverage rate and the moving distance of sensor nodes obtains a higher ranking. The improved formula for calculating the crowding distance entropy is shown in Equation (12).
(12)$$CD\left(i\right)=\frac{\left|f_{1}\left(X_{i+1}\right)-f_{1}\left(X_{i-1}\right)\right|}{\underset{1\leq j\leq h}{\max}f_{1}\left(X_{j}\right)-\underset{1\leq j\leq h}{\min}f_{1}\left(X_{j}\right)}+\frac{\left|f_{2}\left(X_{i+1}\right)-f_{2}\left(X_{i-1}\right)\right|}{\underset{1\leq j\leq h}{\max}f_{2}\left(X_{j}\right)-\underset{1\leq j\leq h}{\min}f_{2}\left(X_{j}\right)}\;+\frac{\underset{i\leq j\leq h}{\min}\left|f_{1}\left(X_{j}\right)-f_{2}\left(X_{j}\right)\right|}{\left|f_{1}\left(X_{i}\right)-f_{2}\left(X_{i}\right)\right|}\;$$
where *h* is the number of individuals in this non-dominated front. *J* is an individual in the same non-dominated fronts with the point *i*.

After determining the non-dominated rank of each solution, the crowding distance entropy of each solution is calculated. The solutions are ranked according to their non-dominated rank and crowding distance entropy in the following way: for two individuals with different non-dominated ranks, the one with a lower non-dominated rank is preferred over the one with a higher non-dominated rank; for individuals with the same dominance rank, the one with larger crowding distance entropy is preferred over the one with smaller crowding distance entropy.

The explorers are set to be the top *PD* individuals sorted by the non-dominated sorting algorithm, and the rest of the individuals become the followers. In the standard SSA algorithm, the number of explorers is generally set to a constant value between 10% and 20% of the total population size, which cannot maximize the role of explorers in the whole population. At the beginning of the algorithm, the global search is more demanding, and more explorers are needed to explore the global better position to improve the global search capability. In the middle and late stages of the algorithm operation, a small number of explorers can both ensure a certain global exploration capability and enhance the local optimization seeking capability. Therefore, the ratio of the number of explorers to the total population size is set as the dynamic weighting factor in this paper, $PD=0.2*\frac{e^{\alpha }}{e^{\alpha }+1}$, and $\alpha =\frac{T}{t}-1$. Where T is the maximum number of iterations, and t is the current number of iterations. With the above equation, it can be realized that at the beginning of the algorithm, the number of explorers is 20% of the entire population, and it gradually decreases to 10% as the algorithm is iterated.

### 5.3. Two-Sample Learning Strategy

In standard SSA, the follower determines its own update method based on the global highest and the global lowest fitness values. The purpose of this update method is to find a random position near the current optimal position. In the other case, if the follower is closer to the global worst position, it is necessary to move to another position. This single way of learning of the current global best or worst position can improve the search speed of the algorithm to some extent, but it also means some important position information is lost and increases the likelihood of the algorithm falling into a local optimum [31].

To enable the followers to better learn the location information of the optimized explorer population, this paper improves the position update formula of the followers in the standard SSA algorithm by using a two-sample learning strategy. As shown in Figure 4, whether the follower *i* learns from the globally optimal individual n, or from the globally suboptimal individual m, it leads to a reduction in diversity.

In order to enable the follower *i* to better search the target location, the current global optimal location and the global suboptimal location are used as guides. When the fitness value of follower *i* is low, it needs to leave the worse position. Additionally, to avoid searching other worse positions again, the global worst position and the global second worst position can be considered in the location update. After improving the update method of the followers by two-sample learning strategy, the followers can make full use of the information of the explorers’ position. The update formula of the positions of the followers can be described as follows:(13)$$X_{i,j}^{t+1}=\{\begin{array}{c} Q\cdot exp\left(r_{1}\frac{X_{w}^{t}-X_{i,j}^{t}}{i^{2}}+r_{2}\frac{X_{z}^{t}-X_{i,j}^{t}}{i^{2}}\right),i>\frac{N}{2}\\ X_{p}^{t+1}+\left[r_{3}\left|X_{p}^{t}-X_{i,j}^{t}\right|+r_{4}\left|X_{m}^{t}-X_{i,j}^{t}\right|\right]\cdot A^{+}\cdot L,otherwise \end{array}$$ where $X_{p}^{t}$ is the optimal position in the current sparrow population. $X_{m}^{t}$ is the next best position in the current sparrow population. $X_{w}^{t}$ is the worst position in the current sparrow population.$\;X_{Z}^{t}$ is the second worst position in the current sparrow population. $r_{1},\;r_{2},\;r_{3},\;r_{4}$ are the step control parameter, and $r_{1}+r_{2}=1,r_{3}+r_{4}=1$.

### 5.4. Procedure for the IM-DTSSA Algorithm

The procedure of IM-DYSSA algorithm is shown in Figure 5, and the steps of the IM-DTSSA algorithm are described as follows:

Step 1: Initialize the parameters of the IM-DTSSA algorithm: population size *N*, the number of sensor nodes n, proportion of explorers *PD*, maximum number of iterations *Itermax*, Safety value *ST*, and *r_1_, r_2_, r_3_, r_4_*.

Step 2: Determine the size of monitoring area, and the nodes are randomly deployed in the monitoring area.

Step 3: Build the triangular network and calculate the value of *Cl* according to Equation (9). Choose the top *N* individuals as the initial population of the algorithm.

Step 4: Sort the individuals by the non-dominated sorting algorithm and calculate the crowding distance entropy according to Equation (11). Choose the top *PD* individuals as the explorer population. Update the explorer location according to Equation (6).

Step 5: Update the position of the followers according to Equation (12).

Step 6: If some sparrows feel the danger, the position of the individuals is updated according to Equation (8).

Step 7: Determine whether the algorithm reaches the maximum number of iterations; if so, output the global solution and record the optimal coverage rate and the moving distance of sensor nodes. Otherwise, skip to step 4 for the next iteration.

### 5.5. IM-DTSSA Time Complexity Analysis

In this section, we analyze the time complexity of our proposed IM-DTSSA algorithm. Suppose the maximum number of iterations of the algorithm is *T*, the population size is *N*, the number of nodes in the network is *n*. Firstly, Delaunay triangulation is performed on the static nodes, and the value of *Cl* for sorting is calculated. Then, the top *N* individuals are used as the initial population of the algorithm. Secondly, the population is sorted by a non-dominated ranking algorithm and the crowding distance entropy is calculated. Finally, the followers’ position according to the two-sample learning strategy is updated. Therefore, we analyze the time complexity of the three steps:(1)The time complexity of Delaunay triangulation is O(*nlogn*), the time complexity of calculating the value of *Cl* is O(*nlogn*), and the time complexity of population initialization is O(*N*).(2)The time complexity of computing the non-dominated rank is O(2*N*^2^), and the time complexity of calculating the crowding distance entropy is O(*NlogN*).(3)The location update of the sparrow population has a time complexity of O(*N*).

Therefore, after introducing the above three improvement strategies, the time complexity of the IN-DTSSA algorithm is O(*TN*^2^). Although the IM-DTSSA algorithm adds a certain amount of computation, the coverage rate and moving distance of nodes of this algorithm are better than other algorithms. 

## 6. Simulation Experiments and Analysis

### 6.1. Experimental Design

To verify the effect of the IM-DTSSA algorithm for WSN coverage optimization, the IM-DTSSA algorithm is compared with DPSO [11], ESSA [19], and NESSA [32]. We designed three sets of experiments to verify the stability of the algorithm performance. To make the experimental data more convincing, the simulation experiments for each algorithm were performed 30 times independently, and the average value was taken as the data for comparison. The simulation experiments were conducted in MATLAB2021a.

In each set of experiments, the experimental environment, such as the number of mobile nodes and static nodes, the sensing radius, and the maximum number of iterations, is the same for all four algorithms. In this paper, the values of parameters of the IM-DTSSA are as follows: φ=ω=0.5, and the values of $r_{1},\;r_{2},\;r_{3},\;r_{4}$ are 0.7, 0.3, 0.7, 0.3, respectively.

### 6.2. Analysis of Simulation Results

#### 6.2.1. Analysis of Coverage Optimization Results

Optimization comparison in the area of 100 m × 100 m

In this section, the sensor nodes were deployed in a square monitoring area of 100 m × 100 m, and the sensing radius of the sensor nodes was 10 m. The maximum iterations was 500, and the number of static nodes and mobile nodes was 30 and 20, respectively. The initial deployment at a static node number of 30 and a dynamic node number of 20 is provided in Figure 6a. The red dots in Figure 6a represent the mobile nodes, and the black “+” signs indicate the static nodes. The circles indicate the sensing area of the nodes. Figure 6b shows the final deployment of sensor nodes, and it can be clearly seen that the mobile nodes are mostly deployed in the uncovered areas of the static nodes.

In addition, as seen in Figure 6c, the IM-DTSSA algorithm optimizes the network coverage rate from the initial 59.60% to 95.63%, achieving an incremental network coverage of 36.03%. The coverage rate of the NESSA algorithm gradually increases and finally reaches 92.21%, but the coverage rate in the initial stage is lower than that of the ESSA algorithm and IM-DTSSA algorithm. The ESSA algorithm achieves an 89.32% coverage rate when running up to 40 generations but falls into a local optimum in the next 200 iterations. It jumps out of the local optimum at the 240th generation and finally reaches a coverage of 90.59%. The coverage rate of the DPSO algorithm increases but finally only reaches a coverage rate of 88.89%. In terms of convergence speed, the DPSO algorithm converges slowly, improving the network coverage rate by 29.91%. The ESSA algorithm reaches the convergence state quickly, but it obviously falls into the local optimal solution, and takes a long time to jump out of the local optimal. Although the NESSA algorithm finally achieves a high coverage rate and does not appear to be significantly trapped in a local optimum, the convergence speed is slow and the growth rate of the coverage rate is slow. The IM-DTSSA algorithm is able to find the better solution quickly from a lower initial coverage and does not appear to be significantly trapped in a local optimum, and it has a strong performance in finding the best solution as well as jumping out of the local optimum.

From Figure 6d and Table 1, it can be seen that the average moving distance of the IM-DTSSA algorithm is the smallest, and the moving distance of the IM-DTSSA algorithm is more uniform. This means the optimization of explorer population has a positive effect on controlling the moving distance of nodes. The remaining energy of each node after deployment is directly related to the moving distance, so the fluctuation size of the distance moved by nodes will directly affect the performance of the WSN. Nodes that move farther away can die sooner, affecting the monitoring effectiveness of the network and even leading to the paralysis of the transmission route. In summary, the IM-DTSSA algorithm proposed in this paper is superior to other algorithms in controlling the moving distance of sensor nodes.

2. Optimization comparison in the area of 50 m × 50 m

In this section, the sensor nodes were deployed in a square monitoring area of 50 m × 50 m, and the sensing radius of the sensor nodes was 5 m. The maximum iterations was 500, and the number of static nodes and mobile nodes was 15 and 20, respectively. The initial deployment at a static node number of 15 and a mobile node number of 20 is shown in Figure 7a. Figure 7b provides the final deployment of sensor nodes.

From Figure 7b, it can be seen that the nodes are more evenly distributed after the IM-DTSSA algorithm network coverage optimization, which greatly improves the redundancy of nodes in the initial stage. Figure 7c shows that the DPSO algorithm improves the coverage rate faster at the beginning, but it starts to fall into a local optimum after the algorithm iterates to 60 generations. Although the algorithm jumps out of the local optimum twice, the increment of coverage was smaller and finally reaches a coverage rate of 83.81%. The coverage rate gradually increases during the coverage optimization by the ESSA algorithm, but finally reached a coverage rate of 89.00% due to the slow increase in coverage rate in the early stage. The coverage rate improved by the NESSA algorithm gradually increases and has a strong ability to jump out of the local optimum, but it still has a certain gap with the IM-DTSSA algorithm in the initial stage. The coverage rate improved by the M-DTSSA algorithm increases from 67.27% to 91.89%, achieving an incremental coverage rate of 24.62%.

Figure 7d and the Table 2 show that the moving distance of the IM-DTSSA algorithm is smaller than the three other algorithms. The fluctuations of the IM-DTSSA algorithm are small, which means the energy of the sensor nodes will be more balanced. At the same time, the moving distance of the DPSO algorithm has the largest fluctuation. The moving distances of the ESSA algorithm and the NESSA algorithm are larger than the moving distance of the IM-DTSSA algorithm.

3.Optimization comparison in the area of 20 m × 20 m

In this section, the sensor nodes were deployed in a square monitoring area of 20 m × 20 m, and the sensing radius of the sensor nodes was 2.5 m. The maximum iterations was 500, and the number of static nodes and mobile nodes was 10 and 15, respectively. The initial deployment at a static node number of 10 and a mobile node number of 15 is shown in Figure 8a. Figure 8b provides the final deployment of sensor nodes.

It is obvious from Figure 8c that the DPSO algorithm reaches a coverage rate of 77% around the 60th generation but was then stuck in a local optimum. The ESSA algorithm always increases the coverage rate, but the speed of increase is relatively slow, reaching a coverage rate of 83.90% around the 40th generation. The NESSA algorithm is close to the IM-DTSSA algorithm in terms of coverage improvement speed, but the IM-DTSSA algorithm achieves higher coverage faster in the early stage. This is because the IM-DTSSA algorithm quickly locates the initial population near the final target area through the Delaunay triangulation strategy. The final optimized network coverage of the IM-DTSSA algorithm increased from the initial 62.48% to 91.84%, achieving a 29.36% increase in coverage rate, which demonstrates that the IM-DTSSA algorithm has a strong global search and the ability to jump out of the local optimum.

As shown from Figure 8d and Table 3, the average moving distance in the area of 20 m × 20 m is smaller than the moving distance in the areas of 50 m × 50 m and 100 m × 100 m. It is still obvious that the moving distance of the IM-DTSSA is smaller than the three other algorithms.

#### 6.2.2. Coverage Comparison with Different Number of Mobile Nodes

In this paper, 30 static nodes were deployed in the area of 100 m × 100 m, and then we recorded the change in coverage when different numbers of mobile nodes were deployed. From Figure 9, it can be seen that when the number of mobile nodes is 10, the coverage rate of the DPSO algorithm and the ESSA algorithm are 64.23% and 73.44%, respectively. The coverage rate of NESSA algorithm is 74.32% and the coverage rate of IM-DTSSA algorithm is 79.36%. When the number of mobile nodes reaches 40, the coverage rate of the DPSO algorithm and the ESSA algorithm are 88.67% and 92.35%, and the coverage rate of ENSSA algorithm is 95.62%. The IM-DTSSA algorithm achieved a coverage rate of 98.89%.

In summary, it can be seen that when the number of mobile nodes is less than 45, the IM-DTSSA algorithm has a better optimized coverage effect. When the number of mobile nodes is greater than 45, the gap between the algorithms gradually decreases, which is due to the fact that all coverage rates can reach more than 85%.

#### 6.2.3. Verification of the Effectiveness of the IM-DTSSA Algorithm

In order to verify the effectiveness of the algorithm, a set of comparison experiments were performed in this study. Each of the three strategies used in this paper were added to the initial algorithm for comparison. In this test, the sensor nodes were deployed in a square monitoring area of 100 m × 100 m, and the sensing radius of the sensor nodes was 10 m. The maximum iterations was 500, and the number of static nodes and mobile nodes was 30 and 20, respectively.

The three strategies used in this paper are the optimization of the initial population, the optimization of explorer population, and the two-sample learning strategy. The SSA algorithm improved by the optimization of the initial population was labeled SSA-1. The SSA algorithm improved by the optimization of the initial population and the optimization of explorer population was labeled SSA-2. According to Table 4 and Figure 10, it can be seen that the three optimization strategies proposed in this paper can significantly improve the performance of the standard sparrow algorithm. It can be found that the algorithms improved by the Delaunay triangulation strategy are all able to improve the coverage of the network in the early stage of the algorithm, as seen in Figure 10a. From Figure 10b, it can be observed that the algorithms improved by the non-dominated sorting algorithm are all able to reduce the moving distance of mobile nodes.

## 7. Conclusions

Aiming at the problems that it is hard to balance the coverage rate of networks and the moving distance of mobile nodes, a multi-strategy improved sparrow search algorithm for WSN coverage optimization (IM-DTSSA) is proposed. Firstly, in the initial stage of the algorithm, the Delaunay triangulation strategy is used to find the location of the uncovered area in the networks, which is filtered and then used as the initial population of the sparrow search algorithm. This approach can quickly move the population to the vicinity of the target area and can reduce the time of the global search of the population in the early stage of the algorithm. Secondly, explorer populations have a significant impact on the performance of SSA algorithm. Therefore, the IM-DTSSA algorithm introduces the idea of non-dominated sorting to optimize the explorer population, thus enhancing the global search capability of the algorithm. Finally, to avoid the problem that the algorithm tends to fall into local optimum, the IM-DTSSA algorithm uses a two-sample learning strategy to improve the follower position update method.

The feasibility and superiority of the IM-DTSSA algorithm for the WSN coverage optimization are demonstrated by comparing the IM-DTSSA algorithm with other three algorithms. The stability of the algorithm is demonstrated by setting different numbers of mobile nodes to observe the changes in the coverage rate of networks. Comparing the average moving distance with the other three algorithms reveals that the IM-DTSSA algorithm is able to reduce the moving distance of mobile nodes while ensuring the coverage rate of WSNs. Our research is aimed at the network environment with a mixture of static and mobile nodes. Firstly, in the initial stage of the IM-DTSSA algorithm, the Delaunay triangulation strategy is used to find the location of the uncovered area in the networks, which is composed of static nodes. Then, the optimal node deployment solution of mobile nodes is found by the next steps. Therefore, the algorithm proposed in this research has a superior performance in dealing with the coverage optimization problem in a network environment composed of hybrid nodes.

## Figures and Tables

**Figure 1 sensors-23-04124-f001:**
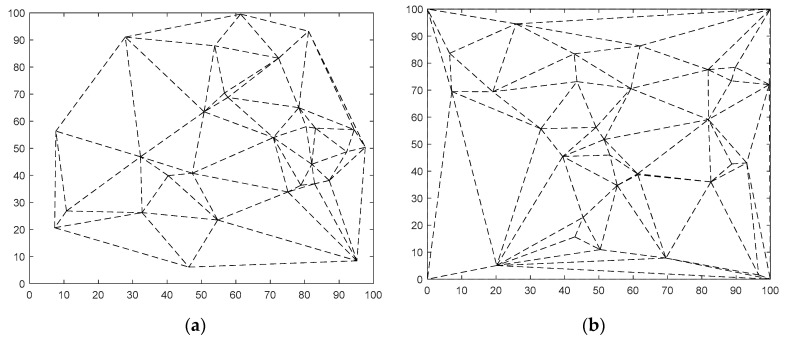
Triangle Network. (**a**) Delaunay triangulation of nodes; (**b**) Regional Delaunay triangulation.

**Figure 2 sensors-23-04124-f002:**
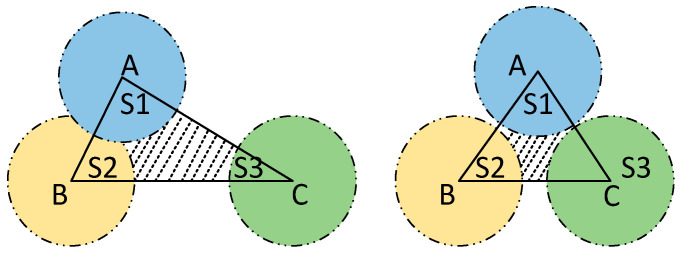
Diagram of coverage vulnerability. The three different colors represent the sensing areas of different nodes.

**Figure 3 sensors-23-04124-f003:**
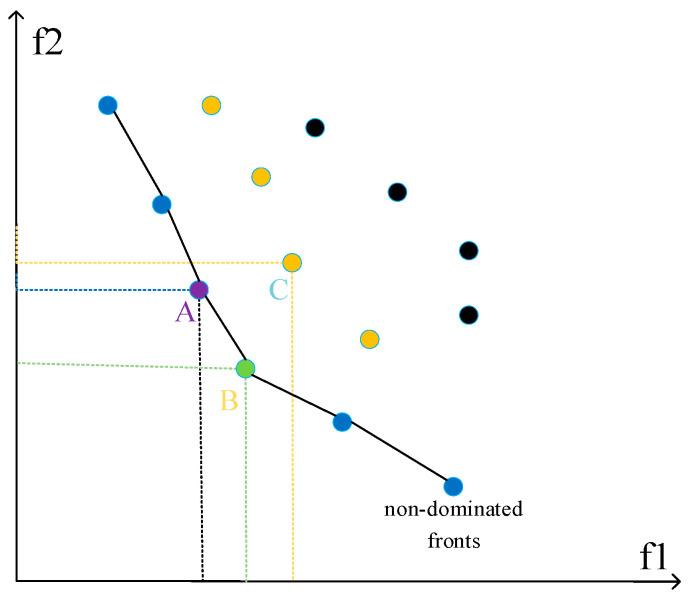
Diagram of non-dominated fronts. The letters A, B and C represent the solution A, B and C. Points of the same color represent solutions in the same dominance fronts.

**Figure 4 sensors-23-04124-f004:**
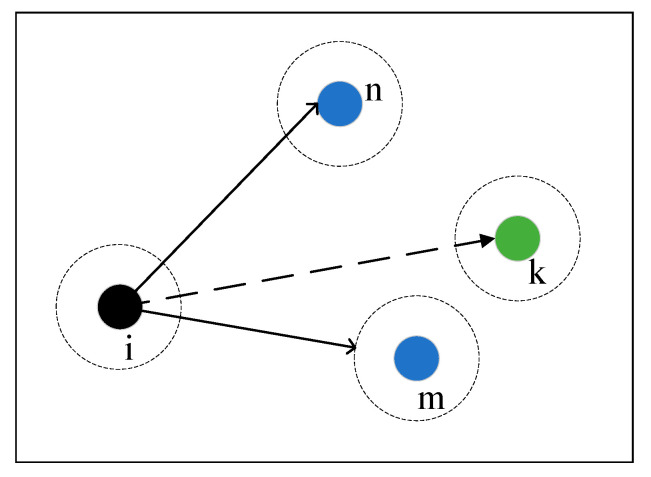
Two-sample learning strategy.

**Figure 5 sensors-23-04124-f005:**
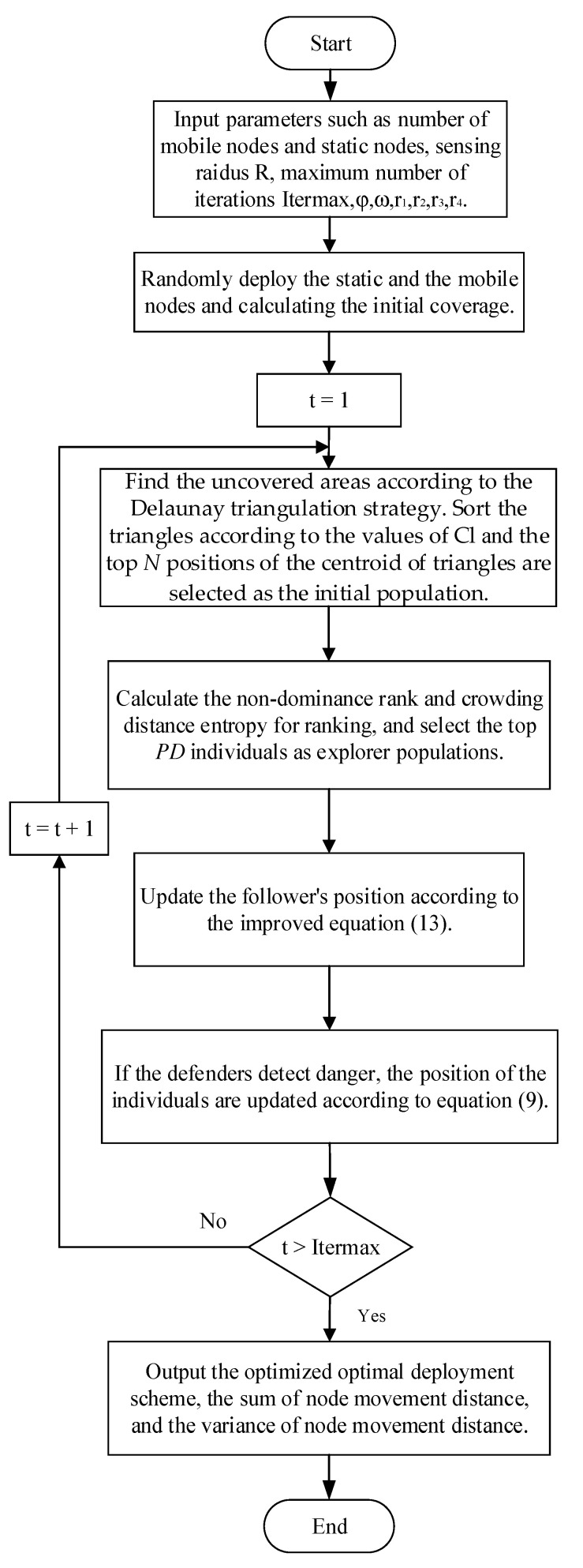
Flowchart of the IM-DTSSA coverage optimization algorithm.

**Figure 6 sensors-23-04124-f006:**
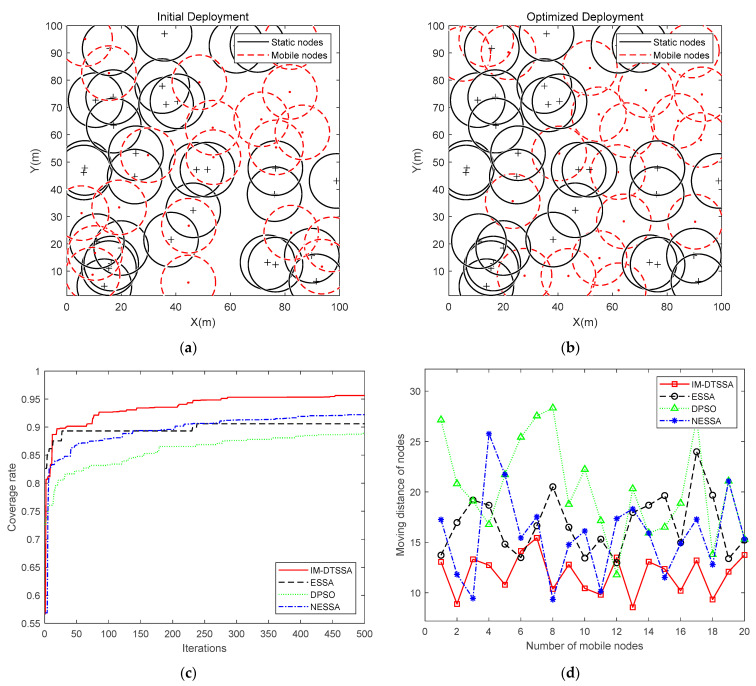
100 m × 100 m coverage optimization comparison: (**a**) Initial Deployment; (**b**) Optimized Deployment; (**c**) Iteration curves of the four algorithms; (**d**) The moving distance of each node.

**Figure 7 sensors-23-04124-f007:**
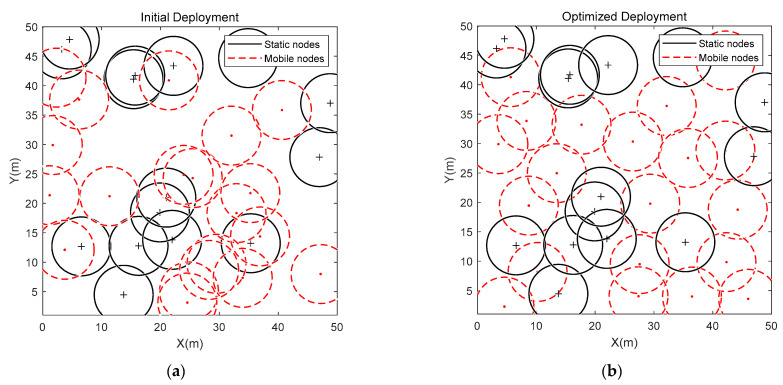
50 m × 50 m coverage optimization comparison: (**a**) Initial Deployment; (**b**) Optimized Deployment; (**c**) Iteration curves of the four algorithms; (**d**)The moving distance of each node.

**Figure 8 sensors-23-04124-f008:**
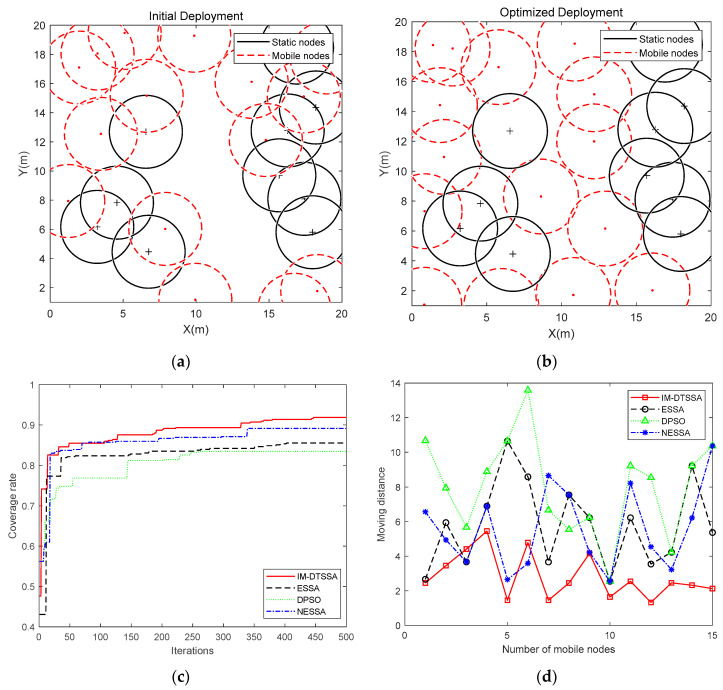
20 m × 20 m coverage optimization comparison: (**a**) Initial Deployment; (**b**) Optimized Deployment; (**c**) Iteration curves of the four algorithms; (**d**)The moving distance of each node.

**Figure 9 sensors-23-04124-f009:**
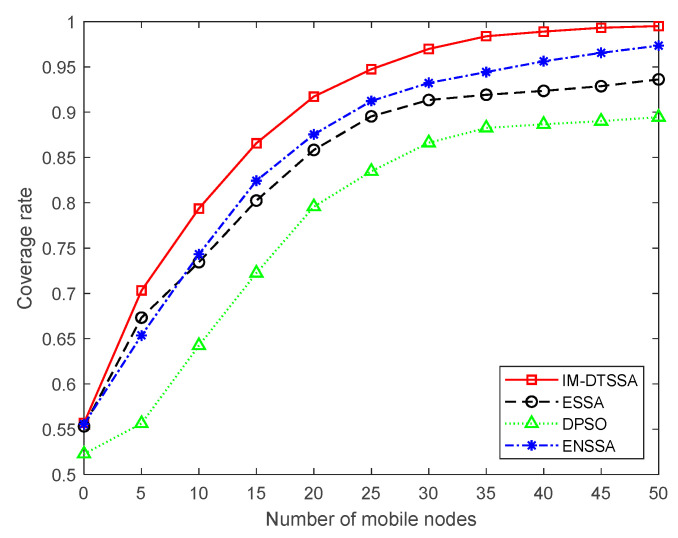
Comparison of algorithm coverage with different number of mobile nodes.

**Figure 10 sensors-23-04124-f010:**
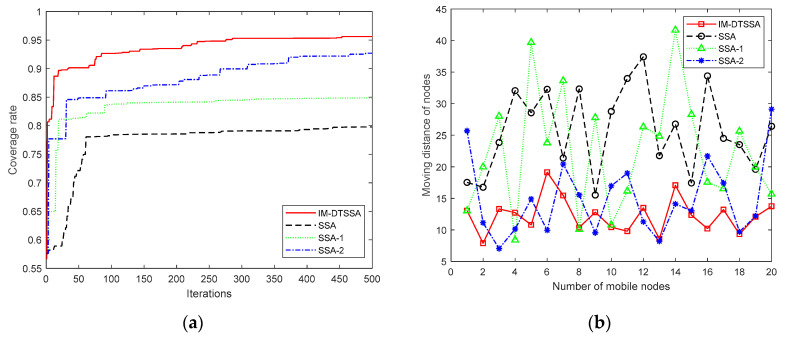
Comparison of the effectiveness of the algorithms.

**Table 1 sensors-23-04124-t001:** Comparison of WSN performance in the area of 100 m × 100 m.

	Number of Static Nodes	Number of Mobile Nodes	Initial Coverage Rate	Optimized Coverage Rate	Average Moving Distance
ESSA	30	20	57.67%	90.59%	16.33 m
DPSO	30	20	58.98%	88.89%	20.29 m
NESSA	30	20	56.72%	92.21%	15.45 m
IM-DTSSA	30	20	59.60%	95.63%	12.36 m

**Table 2 sensors-23-04124-t002:** Comparison of WSN performance in the area of 50 m × 50 m.

	Number of Static Nodes	Number of Mobile Nodes	Initial Coverage Rate	Optimized Coverage Rate	Average Moving Distance
ESSA	15	20	65.29%	89.00%	12.34 m
DPSO	15	20	60.67%	83.81%	15.50 m
NESSA	15	20	67.10%	90.21%	10.27 m
IM-DTSSA	15	20	67.27%	91.89%	6.32 m

**Table 3 sensors-23-04124-t003:** Comparison of WSN performance in the area of 20 m × 20 m.

	Number of Static Nodes	Number of Mobile Nodes	Initial Coverage Rate	Optimized Coverage Rate	Average Moving Distance
ESSA	10	15	63.56	85.53%	5.79 m
DPSO	10	15	65.46	83.45%	7.94 m
NESSA	10	15	60.79	89.12%	5.23 m
IM-DTSSA	10	15	62.48	91.84%	2.99 m

**Table 4 sensors-23-04124-t004:** Comparison of the effectiveness of the algorithms.

Initial Population Optimization	Explorer Population Optimization	Two-Sample Learning Strategy	Optimized Coverage Rate	Average Moving Distance
×	×	×	79.78%	25.74 m
√	×	×	84.86%	22.38 m
√	√	×	92.70%	14.85 m
√	√	√	95.63%	12.36 m

## Data Availability

Not applicable.

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
