# Peer review of "A Multi-Strategy Improved Sparrow Search Algorithm for Coverage Optimization in a WSN"

_sensors, 2023, doi:10.3390/s23084124_

Round 1
Reviewer 1 Report
Dear Authors,
A Multi-strategy Improved Sparrow Search Algorithm for Coverage Optimization in a WSN is the important topics in the recent research area, but I have some potential idea to improve the article in some context:
1. The abstract seems to be ambiguity in the line which is starting it need to be rephrased.
2. While using the Delaunay triangulation method, it is necessary to mention the convergence rate of the searching mechanism.
3. The graphs need to represent the convergence ratio of the population search.
4. The parameter cl in the flowchart is not mentioned anywhere.
5. In line 310, With the above equation, it can be realized that at the beginning of the algorithm, the number of explorers is 20% of the entire population, and gradually decreases 10% as the algorithm is iterated. How the values mentioned are possible?
6. There is no more equation to find out the global best by using the algorithm in line 334.
7. May I know what simulation tool is used for finding the results?
8. Whether your research only support for the mobile node as stated in the conclusion? If not so, change the conclusion parameter
Reviewer 2 Report
1. No quantitative result in the Abstract.
2. What's the main contribution of this hybrid algorithm compared with the existing deployment schemes? What's the main motivation for using multi-strategy-based coverage algorithm?
3. Please separate related works from Introduction. Authors need to outline drawbacks of existing works and add more recent works.
4. Authors need to justify and complete the statement "introduced to improve the quality of the explorer population, and it helps alleviate the problem that the algorithm is difficult to balance multiple objectives" Provide conflict objectives .
5. Authors need to define function ?1 before used it, Equation 11. Is it the fitness function?
6. What is the objective function to obtain the optimal results
7. Justify how could you fix the number of explorers to 20% in formula of PD - Line 380.
8. Please specify the input parameters in the flowchart figure 5;
9. Is objective function in
10. What are values of different parameters ?1, ?2, ?3, ?4
11. Authors need to clarify the use of maximum of iterations of 500 and explain why the number of sensors for fixed nodes and mobile nodes is chosen 30 and 20 ?
12. Can the algorithm find the optimal number of sensors needed for a given deployment area? What will happen to the experimental results?
13. The algorithm is not evaluated for energy consumption and computational complexity.
14. How is the approach different from the existing approaches in performance? Refer to existing optimization algorithms [18 -21] and compare the results.
Reviewer 3 Report
(1) The WSN node coverage model is not detailed, it is not clear what WSN are mainly studied by the author, what is the main object of this model given by the author? It's a generic model, no novelty.
(2) In part 4, what are the problems with the standard sparrow search algorithm? What does the author mainly improve? This part is not clear by comparing the standard.
(3) There is no proper test for verification in this paper. It is only a simulation experiment, and the improvement of the model is not credible. It is suggested that the author add specific experimental test scenes and specific testing process, and how to obtain the data? And so on.
(4) The advantages of the method proposed by the author cannot be reflected in Figure 6 and 7, because some parameters of the simulation are adjustable, and these adjustable parameters are affected by other factors in the actual test, How to determine the changed parameters and the actual basic agreement?
(5) The research topic of this paper is the relatively mature algorithm, and there are many application studies in many literatures. From the whole point of view of the paper, the paper has not fully explained the improvement of the existing algorithm. For example, what is the problem of the traditional method? What problem hasn't it solved yet? What is the main reason? What can be done to improve the problem?
Round 2
Reviewer 2 Report
The authors have satisfactorily responded to all my comments and my decision is accept.
Author Response
Dear reviewer:
Thank you again for your positive comments and valuable suggestions to improve the quality of our manuscript.
Reviewer 3 Report
the overall logic of the paper is reasonable, the current version can be published.
Author Response

(The authors gave the same response as above.)
